# Two Synthetic Peptides Corresponding to the Human Follicle-Stimulating Hormone β-Subunit Promoted Reproductive Functions in Mice

**DOI:** 10.3390/ijms231911735

**Published:** 2022-10-03

**Authors:** Xingfa Han, Xinyu Bai, Huan Yao, Weihao Chen, Fengyan Meng, Xiaohan Cao, Yong Zhuo, Lun Hua, Guixian Bu, Xiaogang Du, Qiuxia Liang, Xianyin Zeng

**Affiliations:** 1Isotope Research Laboratory, Biological Engineering and Application Biology Department, Sichuan Agricultural University, Ya’an 625014, China; 2Key Laboratory for Animal Disease-Resistant Nutrition of the Ministry of Education of China, Animal Nutrition Institute, Sichuan Agricultural University, Ya’an 625014, China

**Keywords:** FSH, synthetic peptide, pharmacological agent, reproductive modulation, mice

## Abstract

A follicle stimulating hormone (FSH) is widely used in the assisted reproduction and a synthetic peptide corresponding to a receptor binding region of the human (h) FSH-β-(34–37) (TRDL) modulated reproduction. Furthermore, a 13-amino acid sequence corresponding to hFSH-β-(37–49) (LVYKDPARPKIQK) was recently identified as the receptor binding site. We hypothesized that the synthetic peptides corresponding to hFSH-β-(37–49) and hFSH-β-(34–49), created by merging hFSH-β-(34–37) and hFSH-β-(37–49), modulate the reproductive functions, with the longer peptide being more biologically active. In male or female prepubertal mice, a single injection of 200 μg/g BW ip of hFSH-β-(37–49) or hFSH-β-(34–49) hastened (*p* < 0.05) puberty, whereas the same treatments given daily for 4 d promoted (*p* < 0.05) the gonadal steroidogenesis and gamete formation. In addition of either peptide to the in vitro cell cultures, promoted (*p* < 0.05) the proliferation of primary murine granulosa cells and the estradiol production by upregulating the expression of *Ccnd2* and *Cyp19a1*, respectively. In adult female mice, 200 μg/g BW ip of either peptide during diestrus antagonized the FSH-stimulated estradiol increase and uterine weight gain during proestrus. Furthermore, hFSH-β-(34–49) was a more potent (*p* < 0.05) reproductive modulator than hFSH-β-(37–49), both in vivo and in vitro. We concluded that hFSH-β-(37–49) and especially hFSH-β-(34–49), have the potential for reproductive modulation.

## 1. Introduction

A follicle-stimulating hormone (FSH), synthesized and secreted by the anterior pituitary, is a heterodimeric glycoprotein composed of an α subunit, which is shared with the glycoprotein hormone family, and a β subunit that confers the hormone receptor specificity [1]. In females, the FSH activates a plasma membrane receptor FSHR on the granulosa cells to promote their proliferation plus ovarian follicular growth, and estradiol biosynthesis [2,3,4]. In males, the FSH binds to FSHR on the Sertoli cells and regulates their proliferation and spermatogenesis [5,6,7]. The exogenous FSH is widely used for the treatment of anovulation and in assisted reproduction technologies [8,9,10]. The exogenous FSH is currently either purified from human urine or the recombinant FSH (rFSH) is produced by recombinant DNA technology [8], although both sources have several limitations [11,12,13]. Therefore, there is an impetus to develop functional alternatives.

The interaction of the FSH with its receptor involves multiple, discrete binding determinants located in the β subunit. Two regions corresponding to the human (h) FSH-β-(33–53) [14] and hFSH-β-(81–95) [15,16] were initially identified as receptor binding sites. Subsequently, a subdomain with four amino acid residues 34–37 (TRDL) [17,18] within hFSH-β-(33–53), and a subdomain with six amino acid residues 90–95 (DSTDCT) [19] within hFSH-β-(81–95) were identified as the core binding determinants of the two larger receptor binding domains, respectively. Recently, hFSH-β-(37–49) (LVYKDPARPKIQK), a 13-amino acid peptide included within a previously reported larger FSH receptor binding domain, hFSH-β-(33–53) [14], was determined to be the receptor binding site [20,21]. Furthermore, the 13-amino acid peptides of the hFSH β-subunit are conserved across mammals [22], and vaccines developed with this 13-amino acid peptide prevented the high FSH-induced obesity in ovariectomized mice [22] and caused subfertility of both male (unpublished data) and female [23] mice.

We hypothesized that the synthetic peptides corresponding to hFSH-β-(37–49) and hFSH-β-(34–49), created by merging hFSH-β-(34–37) and hFSH-β-(37–49), modulate the reproductive functions, with the longer peptide being more biologically active. In the present study, we synthesized these two peptides and tested their bioactivities both in vivo and in vitro.

## 2. Result

### 2.1. FSHβ13AA/FSHβ16AA Accelerated the Onset of Puberty in Both Sexes

Compared to a vehicle-injected control, a single injection of FSHβ13AA or FSHβ16AA (Figure 1A) on day 25, advanced the pubertal onset in male mice by 3–5 d (*p* < 0.05; Figure 1B), with FSHβ16AA resulting in the pubertal onset, on average, 3 d earlier than in FSHβ13AA-injected mice (*p* < 0.05; Figure 1B). There was no significant difference between these two groups for body weight at the pubertal onset, but both reached the pubertal onset at a lower body weight than the vehicle-injected controls (*p* < 0.05; Figure 1C).

A single treatment with either FSHβ13AA or FSHβ16AA on day 25 accelerated the pubertal onset in female mice, with FSHβ16AA being more efficacious than FSHβ13AA (Figure 1D,F). Treatment with FSHβ13AA had minimal effects on the age of the vaginal opening (Figure 1D), but resulted in the first estrus on average 2 d earlier, when compared to controls (*p* < 0.05; Figure 1F). Neither FSHβ13AA nor FSHβ16AA significantly affected the body weight of prepubertal female mice on the day of the vaginal opening (*p* > 0.05; Figure 1E).

### 2.2. Daily FSHβ13AA/FSHβ16AA Promoted Gonadal Development in Both Sexes

Compared to the vehicle-injected males, neither FSHβ13AA or FSHβ16AA affected (*p* > 0.05) the body weight (Figure 2A) or testes weight (Figure 2B,C) in the prepubertal male mice. In contrast, the serum testosterone concentrations were markedly elevated (*p* < 0.05) in males given either peptide, compared to the controls (Figure 2D). Furthermore, FSHβ13AA or FSHβ16AA both upregulated (*p* < 0.05; Figure 2E) the mRNA expressions of *Star*, the gene encoding the rate-limiting enzyme for steroidogenesis. The exogenous FSHβ13AA tended to upregulate (*p* = 0.09) the mRNA expression of the spermatogenesis-associated gene *Aqp8* in the testes (Figure 2E), whereas FSHβ16AA markedly upregulated (*p* < 0.05; Figure 2E) all of the spermatogenesis-associated genes assayed, including *Creb*, *Klf4*, *Gja1* and *Aqp8* in the testes. On day 29, there was no mature sperm in the testes of the control males (Figure 3A), but those treated with peptides had sperm (Figure 3B,C).

The peptide treatments had no effects on the body, ovary or uteri weight on day 29 in female mice (*p* > 0.05; Figure 4A–D), but increased the serum 17β-estradiol concentrations (*p* < 0.05, Figure 4E). Furthermore, FSHβ13AA upregulated (*p* < 0.05; Figure 4F) the mRNA expressions of the steroidogenesis-associated genes *HSD3β1* and *Cyp19a1*, and the follicle development-associated genes *Amh* and *Ar* in the ovaries. With the exception of these genes, FSHβ16AA upregulated (*p* < 0.05; Figure 4F) the mRNA expressions of the steroidogenesis-associated genes *Star* and *Cyp17a1*, and the follicle development-associated genes *Ccnd2* and *Nobox* in the ovaries and tended (*p* < 0.1) to upregulate the mRNA expressions of *Fshr*, *Foxo3a* and *Bmp15*. In addition, both FSHβ13AA and FSHβ16AA markedly promoted folliculogenesis (Figure 5A–C), as evidenced by more antral follicles (Figure 5D).

### 2.3. FSHβ13AA/FSHβ16AA Promoted the GC Proliferation and the 17β-Estradiol Production In Vitro

Compared to vehicle-treated control, FSHβ13AA or FSHβ16AA increased the 17β-estradiol production in GCs in a dose-dependent manner, with a plateau at 100 ng/mL for each (*p* < 0.05; Figure 6A), but a reduction in the 17β-estradiol (*p* < 0.05; Figure 6A) when the dose of either was increased to 300 ng/mL. In addition, both FSHβ13AA and FSHβ16AA upregulated the mRNA expressions of *Cyp19a1* in GCs from 0 to 100 ng/mL in a dose-dependent manner (*p* < 0.05; Figure 6B), whereas at 300 ng/mL, the expression of *Cyp19a1* was reversed (*p* < 0.05) (Figure 6B).

From 30 to 300 ng/mL of FSHβ13AA or FSHβ16AA promoted the proliferation of GCs in vitro (*p* < 0.05; Figure 7A,B), but only FSHβ16AA had a dose-dependent response (*p* < 0.05; Figure 7C). Similarly, both FSHβ13AA and FSHβ16AA upregulated the *Ccnd2* expression in GCs, but only FSHβ16AA did so in a dose-dependent manner (*p* < 0.05; Figure 7D). Both FSHβ13AA and FSHβ16AA upregulated the FSH target gene *Inhα* expression in GCs, especially at high doses, but only FSHβ16AA exhibited a dose-dependent manner (*p* < 0.05; Figure 7E). Finally, FSHβ16AA was more potent (*p* < 0.05) than FSHβ13AA in upregulating the *Ccnd2* and *Inhα* expressions in GCs, at both low and high doses (30 and 300 ng/mL; Figure 7D,E).

To verify that the effects of FSHβ13AA/FSHβ16AA on the reproductive modulation were not due to their size and/or global physico-chemical properties, a mixed scrambled peptide control (Appendix A) was designed for each peptide. In the experiments above, both FSHβ13AA and FSHβ16AA had the highest activities at 100 ng/mL. Compared to FSHβ13AA/FSHβ16AA, 100 ng/mL of the mixed scrambled peptides did not (*p* > 0.05) promote the 17β-estradiol production nor the cell proliferation of GCs in vitro (Figure 8). Similarly, there was no difference (*p* > 0.05) between the scrambled FSHβ13AA/FSHβ16AA peptides (100 ng/mL) and the vehicle control in promoting the cell proliferation or estrogen production of GCs (Figure 8).

### 2.4. FSHβ13AA/FSHβ16AA Antagonized Endogenous FSH Actions in the Adult Female Mice

During each estrus cycle in mice, along with a rise in the circulating FSH concentrations during proestrus, the FSH increases both the estrogen production and the uterus weight, with both peaking during proestrus [24]. Compared to the vehicle-injected control, a single FSHβ13AA or FSHβ16AA treatment during diestrus mitigated the proestrus-associated increases (*p* < 0.05) in the ovary weight and uterus weight, and in the serum 17β-estradiol concentrations (Figure 9A–C), with no differences between the FSHβ13AA- and FSHβ16AA-injected females (*p* > 0.05; Figure 9A–C). Furthermore, based on qPCR, either FSHβ13AA or FSHβ16AA during diestrus down-regulated the mRNA expressions of the estrogen biosynthesis-associated genes, including *Lhcgr*, *Star*, *Hsd3β1* and *Cyp17a1* in the ovaries during proestrus. Excepting these genes, FSHβ16AA also markedly down-regulated (*p* < 0.05; Figure 9D) the mRNA expressions of the estrogen biosynthesis-associated genes *Cyp11a1* and *Cyp19a1*, and the follicle development-associated gene *Cdh1* in the ovaries. However, these genes only tended (*p* < 0.1) to be downregulated in the FSHβ13AA-injected females, indicating a greater potency of FSHβ16AA than FSHβ13AA for antagonizing the endogenous FSH actions. Finally, FSHβ16AA during diestrus selectively upregulated (*p* < 0.05) the *Amh* expressions in the ovaries during proestrus.

## 3. Discussion

Based on the in vivo and in vitro studies, we reported for the first time that the synthetic peptide corresponding to hFSH-β-(37–49) (LVYKDPARPKIQK, FSHβ13AA), a newly identified receptor-binding site of the hFSH-β subunit [20,21], modulated the reproductive processes, including hastening puberty in male and female mice, and enhancing the ovarian GC proliferation and 17β-estradiol production. In adult female mice, it partially antagonized the endogenous FSH, suppressing the endogenous FSH-stimulated estradiol production and the uterine weight gain during proestrus. These results were in good agreement with previous studies that synthetic peptides corresponding to hFSH-β-(33–53) bound to the FSH receptors, functioned as a partial FSH agonist both in vivo and in vitro, and acted as a partial receptor-binding antagonist of the endogenous FSH in adult females [14]. Thus, similar to other receptor-binding peptide segments of the hFSH β subunit [14,15,16,17,18,19], the synthetic peptide (LVYKDPARPKIQK) corresponding to hFSH-β-(37–49) has promise as a novel class of FSH pharmacological agents, with the potential for reproductive modulation.

There is emerging evidence that the α subunit of the FSH plus complex glycosylation modifications in both α and β subunits are required for the complete expression of the hormone action, although they are not essential for the effective signal transduction [14,15,16,17,18,19]. Furthermore, synthetic peptides corresponding to the functional receptor-binding regions of the hFSH-β subunit are capable of modulating the reproductive processes both in vivo and in vitro [14,15,16,17,18,19]. The hFSH-β-(37–49) is a subdomain within a larger receptor binding region of the hFSH β subunit, i.e., hFSH-β-(33–53) [14]. Within hFSH-β-(33–53), a tetrapeptides (TRDL) subdomain corresponding to hFSH-β-(34–37) is the predominant binding determinant. In that regard, the synthetic peptide representing the larger receptor binding domain hFSH-β-(33–53) modulated the reproductive processes, but only if it had the tetrapeptides (TRDL) subdomain [17]. In addition, the TRDL peptide per se had the same functions as hFSH-β-(33–53) in modulating the reproductive processes both in vivo and in vitro [17,18]. Therefore, we inferred that the TRDL sequence within hFSH-β-(33–53) is a functional site, and pivotal for the post receptor-binding signal transduction. The hFSH-β-(37–49) sequence tested in the present study is located next to the TRDL sequence region in the hFSH β subunit, with one amino acid overlap between the two (see Figure 1A). Given the importance of these two peptide sequences in receptor binding, we hypothesized that hFSH-β-(34–49) (TRDLVYKDPARPKIQK, FSH β16AA), created by merging hFSH-β-(34–37) and hFSH-β-(37–49), would be the most biologically active. As expected, FSHβ16AA (corresponding to hFSH-β-(34–49)) was significantly more potent than hFSH-β-(37–49) (FSHβ13AA) in accelerating the pubertal onset in prepubertal male and female mice, and in promoting the estradiol production and the cell proliferation in cultured mouse ovarian granulosa cells. Without using TRDL in the present study, we do not know its potency compared to hFSH-β-(34–49) (TRDLVYKDPARPKIQK) for the reproductive modulation. However, as hFSH-β-(34–49) without the TRDL sequence modulated reproductive processes in mice (REF), hFSH-β-(34–49) should be a functional receptor-binding site that can solely induce the postbinding signal transduction. Consequently, the synthetic peptide corresponding to hFSH-β-(34–49) should be a very good (perhaps optimal) choice for a novel class of FSH pharmacological agents, with the potential for reproductive modulation and control.

Treatment with FSHβ13AA or FSHβ16AA for 4 d significantly elevated the serum testosterone and the 17β-estradiol concentrations in prepubertal male and female mice, respectively, consistent with a report that a single injection of FSH related synthetic peptides in sexually immature mice increased the concentrations of sex steroids [18]. The ability of FSHβ13AA/FSHβ16AA to promote gonadal steroidogenesis was attributed to their ability to upregulate the expression of key steroidogenic genes, e.g., *Star* in the testes, and *Star*, *Cyp17a1*, *HSD3β1* and *Cyp19a1* in the ovaries of prepubertal mice. Furthermore, both FSHβ13AA and FSHβ16AA significantly upregulated the mRNA expressions of the FSH target gene *Cyp19a1*, and thus increased the 17β-estradiol production in cultured murine primary GCs. However, there was no significant difference between the FSHβ13AA/FSHβ16AA-injected animals and the controls regarding the weights of the testes or ovaries and uteri, which differed from previous reports [18]. Perhaps the interval after the peptide administration was too short to allow for the difference to be manifested, as the gonadal weights were determined immediately after the peptide administration in our study, versus 1 wk later [18].

Treatment with FSHβ13AA/FSHβ16AA daily, from days 25 to 28, promoted spermatogenesis in the prepubertal male mice when assessed on day 29. To the best of our knowledge, this is the first report to give the exogenous FSH-related synthetic peptide in prepubertal male mice that promoted the complete spermatogenesis. Furthermore, these peptides upregulated the mRNA expressions of the key spermatogenesis-associated genes, i.e., *Creb*, *Klf4*, *Aqp8* and *Gja1* in the testes. Of those, *Creb* [25] and *Klf4* [26] have essential roles in integrating and mediating the signaling pathways of the FSH in regulating spermatogenesis. *Gja1* (also known as connexin 43), a prominent testicular gap junction protein, has essential roles in controlling spermatogenesis by regulating the Sertoli cell proliferation [27]. The conditional knockout *Gja1* caused the spermatogenic arrest at the early spermatogonia stage [27]. Particularly, *Aqp8*, encoding aquaporin 8, is required for the differentiation of the spermatids into sperm during spermatogenesis [28]. The upregulation of these key spermatogenesis-associated genes, especially the FSH target transcription factor genes (*Creb* and *Klf4*) confirmed the ability of FSHβ13AA and FSHβ16AA to promote the complete spermatogenesis by activating at least the FSH receptor signaling pathways.

Similarly, four consecutive FSHβ13AA/FSHβ16AA treatments markedly induced folliculogenesis in prepubertal female mice, based on the antral follicle counts. Many key genes involved in the follicle activation and the early follicle development, including *Foxo3a* [29], *Nobox* [30], *Amh* [31] and *Wnt2* [32], as well as the GC proliferation and differentiation including *Ccnd2* [33], *BMP15* [34] and *Ar* [35], were significantly upregulated by FSHβ13AA/FSHβ16AA. Among these upregulated genes, *Wnt2* and *Ccnd2* are both downstream genes of the FSH-FSHR signaling. *Wnt2* integrates multiple pathways in mediating the FSH action on controlling the follicular activation and the early development [32], whereas *Ccnd2* (encoding cyclin D2) is specifically expressed in GCs of growing follicles, and essential for the mediating actions of the FSH on the GC proliferation and follicular growth [33]. Deletion of *Cyclin D2* impairs the GC proliferation and prevents follicle development [33]. The upregulation of *Ccnd2* by FSHβ13AA/FSHβ16AA was also directly validated by the in vitro culture of GCs. The increased expression of these key genes, especially the FSH target genes *Wnt2* and *Ccnd2*, accounted for FSHβ13AA and FSHβ16AA promoting folliculogenesis. Furthermore, FSHβ13AA/FSHβ16AA tended to upregulate the expression of *Fshr*, a FSH target gene [36], with a potential role in the antrum formation in prepubertal mice [37].

Giving either FSHβ13AA or FSHβ16AA during diestrus to adult mice significantly antagonized the FSH-stimulated estradiol synthesis and increased the ovary and uterus weights during proestrus. Similar outcomes were reported for other synthetic FSH related peptides [14,17]. The ovarian gene expression also supported the interpretation that FSHβ13AA and FSHβ16AA acted as a partial antagonist of the endogenous FSH, especially suppressing the estradiol biosynthesis. Of all the genes assessed in the ovaries, *Amh* was the only one significantly upregulated by giving FSHβ16AA during diestrus. Regardless, this does not invalidate the FSHβ16AA functioning as a partial antagonist of the endogenous FSH. The anti-müllerian hormone (AMH) is primarily produced by GCs of the pre-antral and small antral follicles [31]. Although the FSH may induce the pre-antral follicles to produce the AMH by promoting the GC proliferation [35], the AMH concentrations are very low during diestrus [38] when FSHβ16AA was given. Therefore, the increased expression of *Amh* in the ovaries during proestrus was attributed to the promoting effects of FSHβ16AA on the GC proliferation in the pre-antral follicles and/or small antral follicles during diestrus immediately after treatment. In support of this concept, in the present study, FHβ16AA promoted the cultured prepubertal mouse primary GCs proliferation in a dose-dependent manner and upregulated the *Amh* in the ovaries of prepubertal mice.

The synthetic FSH related peptides, as a novel class of pharmacological agents, acting as either FSH agonists or antagonists, will be of enormous benefit for human and veterinary use. For example, the increased FSH due to ovarian failure has been implicated in osteoporosis, adiposity, and dyslipidemia in menopausal women [39], as well as in postmenopausal atherosclerosis [40]. Perhaps the synthetic FSH related peptides acting as FSH antagonists would be beneficial for these conditions. Solid-phase peptide synthesis technology has been largely improved, with good yields and purities [41]. Moreover, the synthesis of such short peptides would be cheaper than the isolation of the FSH from the urine of menopausal women or from the production by the recombinant DNA.

How the small hFSH-β subunit peptides, such as hFSH-β-(34–49) and hFSH-β-(37–49), exert their agonist and partial antagonist activities is unknown. Using mixed scrambled peptides as controls excluded the possibility that the responses were due to their size and/or global physico-chemical properties. Stimulation by the FSH requires not only binding to the FSHR extracellular domain (ECD) but also contacts with the hinge region and the 7-transmembrane domain region, responsible for the signal transduction across the plasma membrane [42]. Perhaps the small hFSH-β subunit peptides, such as hFSH-β-(34–49) and hFSH-β-(37–49) act as antagonists on the FSH orthosteric site in the FSHR-ECD and exert their agonistic effects on the allosteric site (s) in the transmembrane region, but this awaits confirmation.

In summary, synthetic peptides corresponding to receptor-binding sites of both hFSH-β-(34–49) and hFSH-β-(37–49), accelerated the pubertal onset in prepubertal male and female mice, promoted the GC proliferation and the 17β-estradiol production in vitro, and partially antagonized the endogenous FSH in adult female mice. The hFSH-β-(34–49) peptide was significantly more biologically active than hFSH-β-(37–49) in modulating the reproductive processes both in vivo and in vitro. Therefore, synthetic peptides corresponding to hFSH-β-(37–49) and especially hFSH-β-(34–49) have the potential for the development of a novel class of FSH pharmacological agents for the reproductive modulation, as well as the treatment of the FSH-caused disorders in humans and animals.

## 4. Materials and Methods

### 4.1. Peptide Synthesis, Purification and Characterization

The synthesis and purification of peptides were carried out as described [22]. In brief, the peptide synthesis of FSHβ13AA (LVYKDPARPKIQK) and FSHβ16AA (TRDLVYKDPARPKIQK) was performed on an ABI 430A peptide synthesizer using 2-chlorotrityl chloride resin (1.0 mmol/g) following a standard Fmoc-based solid phase peptide synthesis strategy. The resulting FSHβ3AA and FSHβ16AA were purified through a reverse-phase high-performance liquid chromatography (HPLC) using a Prep 2025 Column (20 mm × 250 mm, Bischoff, Leonberg, Germany) filled with Polygosil C18 material. According to the peak area, the purity of FSHβ13AA and FSHβ16AA was equal to 98.58 and 97.79%, respectively (Appendix A). The sequence integrity of the two synthesized peptides was validated through mass spectrometry.

The mixed scrambled control peptides for FSHβ13AA/FSHβ16AA were generated at the Mimotopes website (http://www.mimotopes.com) at 21 May 2022 and prepared as described above. The amino acid composition and the mass percentage for each scrambled peptide are in Appendix A. 

### 4.2. Mice and Experimental Design

Healthy male and female C57BL/6J mice (Sichuan University Animal Center, Chengdu, China) were bred to generate progeny for all experiments. The mice were kept in 12-h light/dark cycles at 25 ± 0.5 °C and 50–60% humidity, with ad libitum access to a standard chow and water. The whole experimental design was shown in Figure 10, and the details of each single experiment were described as follows.

#### 4.2.1. Experiment 1: Effects of FSHβ13AA/FSHβ16AA on the Pubertal Onset

Littermate mouse pups (33 males and 56 females) were randomly allocated at weaning (21 d) into two groups, vehicle or treatment. Twelve male pups were treated with the vehicle (0.2 mL sterile 0.9% saline), whereas 10 and 11 male pups were treated with FSHβ13AA or FSHβ16AA (200 μg/g BW in 0.2-mL 0.9% saline), respectively. Similarly, 18 female pups were treated with the vehicle (0.2 mL sterile 0.9% saline) and 18 and 20 female pups were treated with FSHβ13AA or FSHβ16AA (200 μg/g BW in 0.2-mL 0.9% saline ip on day 25). The pubertal onset was assessed by the age of the vaginal opening and the first estrus for females, and by the age of the balanopreputial separation (BPS) for the males, based on daily examinations starting from day 25, as described [43]. For females, once the vaginal opening had occurred, vaginal cytology was assessed each morning between 900 and 1000 h to determine the day of first estrus.

#### 4.2.2. Experiment 2: Effects of FSHβ13AA/FSHβ16AA on the Gonadal Functions in the Prepubertal Mice

To assess the bioactivity of FSHβ13AA/FSHβ16AA in vivo, male and female littermate pups with similar body weight were allocated randomly at weaning into three groups as described above. For males, 11 pups were given 0.2 mL 0.9% saline ip, whereas 12 and 12 pups were given 200 μg/g BW FSHβ13AA or FSHβ16AA in 0.2 mL saline ip from days 25 to 28 (four treatments), respectively. The same schedule was used for the females, with 12 pups given 0.9% saline, and 11 and 12 given FSHβ13AA or FSHβ16AA, respectively. On day 29, all pups were anesthetized with isoflurane (Fluriso; VetOne, ID, USA) and euthanized. The trunk blood was collected and centrifuged at 2000× *g* for 15 min at 4 °C and the sera were stored at −20 °C, pending analyses. For the males, both testes were excised, dissected free of epididymides and weighed as a pair. The right testis was immediately frozen in liquid nitrogen and stored at −80 °C for qPCR and the left testis was fixed in 10% buffered formalin for histology. For the females, the ovaries and uteri were collected and weighed, the right ovary was immediately frozen in liquid nitrogen and stored at −80 °C for the qPCR analysis of the gene expressions, and the left ovary was fixed in 10% buffered formalin for histology.

#### 4.2.3. Experiment 3: Effects of FSHβ13AA/FSHβ16AA on the Granulosa Cell Function and Proliferation In Vitro

Primary granulosa cells were isolated from the ovarian follicles and cultured as described [44], with minor modifications. Briefly, the ovaries were excised from the prepubertal mice at 25 d of age, and individually transferred into 35-mm Petri dishes containing PBS. The GCs were collected by follicle puncture, with the assistance of a surgical dissecting microscope (Olympus, SZ51, Tokyo, Japan). The cells were cultured in DMEM/F12 (HyClone; GE Healthcare Life Sciences, Logan, UT, USA) with 10% fetal bovine serum (Gibco; Thermo Fisher Scientific, Inc., Waltham, MA, USA), 1% penicillin-streptomycin solution (Sangon Biotech Co., Ltd, Shanghai, China) and maintained at 37 °C in a humidified atmosphere containing 5% CO_2_.

For evaluating the effects of FSHβ13AA/FSHβ16AAT on the estradiol production, 3 × 10^4^ cells per well were cultured under standard conditions for 24 h. Then, 0, 30, 100 and 300 ng/mL of FSHβ13AA or SHβ16AA was added and incubated for 24 h. Following the treatment, the cell culture supernatants and the cells in each well were collected and kept at −80 °C, pending further analysis.

Following the culture of 5000 cells per well under standard conditions for 24 h, the proliferation of GCs was determined with a CCK-8 assay. For this, a 10 µL CCK-8 reagent (Solarbio Science & Technology Co., Ltd, Beijing, China) was added to each well, followed by incubation at 37 °C for 0, 6, 12, 18, 24, 30 or 36 h. At each time-point, the cell proliferation was determined by measuring the absorbance (450 nm) with a plate reader (Bio-Rad, Hercules, CA, USA).

To validate the functional specificity of FSHβ13AA/FSHβ16AAT in modulating the reproductive regulation, the in vitro assessment of the ability of FSHβ13AA/FSHβ16AAT to promote the GC proliferation and the estradiol production were repeated exactly as described above, using the optimal dose of FSHβ13AA/FSHβ16AAT, as determined above. Both the vehicle (0.9% saline) and the mixed FSHβ13AA/FSHβ16AAT scrambled peptides (same dose as FSHβ13AA/FSHβ16AAT) were used as the controls in these experiments.

#### 4.2.4. Experiment 4: Effects of FSHβ13AA/FSHβ16AA on the Endogenous FSH Action in Adult Females

Thirty adult females (8–12 wk) were allocated into one of three groups (n = 10): the vehicle, FSHβ13AA or FSHβ16AA, as described above. The systemic FSH concentrations in mice are very low during metestrus and diestrus, rise during proestrus and peak during estrus [38]. Adult female mice were given 200 μg/g BW FSHβ13AA or FSHβ16AA ip in 0.2 mL sterile saline during diestrus, whereas the control mice received 0.2 mL 0.9% saline during diestrus. The estrous cycle stage was detected by vaginal cytology [45] between 0800 and 0900 h each morning. During the first proestrus after treatment, the mice were anesthetized with isoflurane and euthanized. The trunk blood was collected, centrifuged at 2000× *g* for 15 min at 4 °C and the sera stored at −20 °C. The ovaries and uteri were collected and weighed and the ovaries were immediately frozen in liquid nitrogen and stored at −80 °C for the subsequent qPCR.

#### 4.2.5. Serum and Cell Culture Media Hormones Assays

Concentrations of 17β-estradiol (KGE014) in the serum of the female mice and mouse primary ovarian granulosa cell culture media, and the serum testosterone (KGE010) concentrations in the male mice were assayed using the commercially available enzyme-linked immunosorbent assay (ELISA) kits, according to the manufacturer’s instructions (R&D Systems; Bio-Techne, Minneapolis, MN, USA). Each sample was assayed in duplicate. Sensitivity was 12.1 pg/mL for 17β-estradiol, and 0.041 ng/mL for testosterone. The intra- and inter-assay CVs were lower than 5.6% and 7.8%, respectively for 17β-estradiol, and lower than 4.3% and 6.9%, respectively for testosterone.

#### 4.2.6. Gonadal Histology

Following the fixation in 10% buffered formalin, the ovarian and testicular tissues were routinely processed for the histological analysis, including paraffin embedding, sectioning (4 μm), mounting on glass slides, and staining with hematoxylin and eosin (H&E), prior to the examination with a light microscope. The quantification of antral follicle numbers in the ovaries was performed in every fifth section for a total of 10 sections per ovary, as described [43].

#### 4.2.7. Quantitative Analysis of the mRNA Expression

The total RNA was isolated from the ovaries, testes, and cultured ovarian granulosa cells using a commercial kit (Invitrogen Co., Carlsbad, CA, USA), according to manufacturer’s instructions. The ratio of the absorbance at 260 and 280 nm was determined and the agarose gel electrophoresis was carried out. A total of 500 ng RNA was converted into first-strand cDNA using a PrimeScript^®^ RT reagent kit with gDNA Eraser (TaKaRa Bio, Co., LTD, Dalian, China). The quantitative real-time PCR was carried out on a CFX96 Real Time PCR detection system (Bio Rad, Hercules, Calif, USA). The PCR reaction contained 1 μL cDNA, 500 nmol/L each of the forward and reverse primers, and 2×SYBR^®^ premix TaqTM (TaKaRa Bio Co., Ltd, Dalian, China). The primer sequences of the target and reference genes are shown (Table 1). The PCR cycling conditions were: initial denaturation at 95 °C (1 min), followed by 40 cycles of denaturation at 95 °C (5 s), annealing at 60 °C (25 s) and a final melting curve analysis (to monitor the PCR product purity). A reference house-keeping gene (18 s) was measured for each sample. The fold change of the mRNA in the treatment group relative to the control was determined by 2^−ΔΔCt^.

#### 4.2.8. Statistical Analyses

The comparisons among the groups were accomplished by one-way ANOVA, followed by Tukey’s test. For the analysis of the repeated measures (i.e., GC proliferation), a two-way ANOVA followed by Sidak’s multiple comparisons test was used. All analyses were carried out with GraphPad Prism 9.2 (La Jolla, CA, USA), with the significance defined as *p* < 0.05 and values expressed as mean ± SEM.

## Figures and Tables

**Figure 1 ijms-23-11735-f001:**
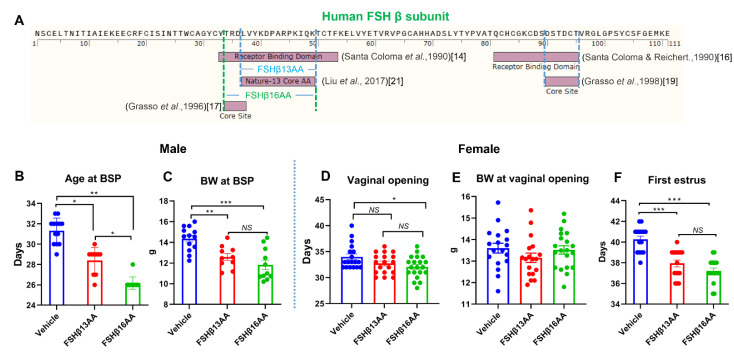
FSHβ13AA/FSHβ16AA hastens puberty in male and female mice. (**A**) Timeline in the identification of the receptor-binding sites of human (h) FSH β subunit. In 1990, two regions corresponding to hFSH-β-(33–53) [14] and hFSH-β-(81–95) [16] were identified as receptor binding sites. Later, within the two larger receptor-binding domains of the hFSH-β-subunit, a subdomain with four amino acid residues 34–37 (TRDL) [17] within hFSH-β-(33–53), and another subdomain with six amino acid residues 90–95 (DSTDCT) [19] within hFSH-β-(81–95) were identified as core binding determinants. In 2017, Liu et al. reported the receptor binding site of the hFSH-β-subunit was a 13-amino acid sequence (33–53), referred to as Nature-13 core AA, as this discovery was published in Nature [21]. (**B**) A single treatment (200 μg/g BW) of either FSHβ13AA or FSHβ16AA (**A**) on day 25 accelerated the pubertal onset in male mice as assessed by balanopreputial separation (BPS). (**C**) Body weight on the day of the pubertal onset in male mice. (**D**) One treatment (200 μg/g BW) of FSHβ13AA/FSHβ16AA (**A**) on day 25 hastened puberty in females as assessed by the vaginal opening. (**E**) Body weight on the day of the vaginal opening. (**F**) Similarly, these peptide treatments (**A**) on day 25 accelerated the pubertal onset (based on first estrus). Data are expressed as mean ± SEM. *NS*, not significant, * *p* < 0.05, ** *p* < 0.01, *** *p* < 0.001.

**Figure 2 ijms-23-11735-f002:**
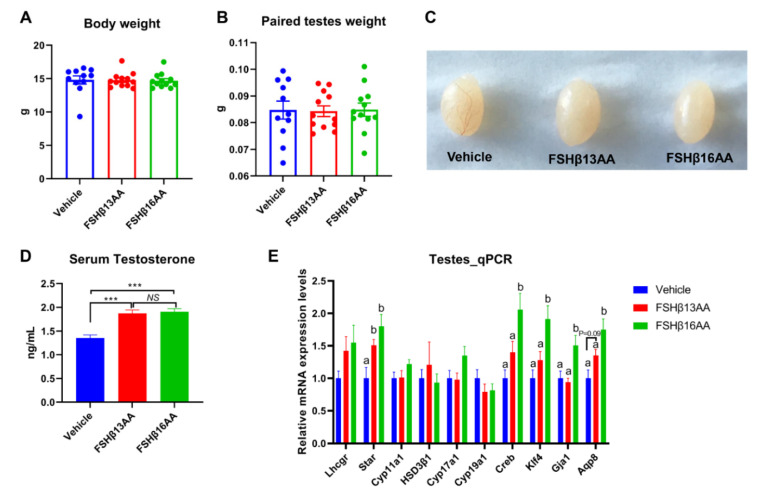
Repeated FSHβ13AA/FSHβ16AA treatments promote testicular steroidogenesis in prepubertal male mice. Giving 200 μg/g BW of either FSHβ13AA or FSHβ16AA (Figure 1A) once daily on days 25 to 28 had no significant effects on the body (**A**) or the paired testes weights (**B**) of the prepubertal male mice on day 29. (**C**) Representative testes in all of the treatment groups. (**D**) Serum testosterone concentrations of the male mice on day 29. (**E**) Levels of the mRNA expression of the steroidogenesis- and spermatogenesis-associated genes in the testes on day 29. *Lhcgr*, luteinizing hormone receptor; *Star*, Steroidogenic acute regulatory protein; *Cyp11a1*, cytochrome P450 family 11 subfamily A member 1; *Hsd3β1*, 3beta-hydroxysteroid dehydrogenase type 1; *Cyp17a1*, cytochrome P450 family 17 subfamily A member 1; *Cyp19a1*, cytochrome P450 family 19 subfamily A member 1 (aromatase); *Creb*, cAMP response element-binding protein; *Klf4*, Kruppel-like factor 4; *Gja1*, Gap junction alpha-1 protein; *Aqp8*, aquaporin 8; Data are mean ± SEM. ^a,b^ Means without a common letter differed (*p* < 0.05), *NS*, not significant, *** *p* < 0.001.

**Figure 3 ijms-23-11735-f003:**
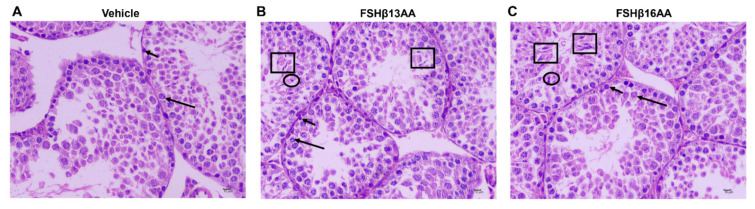
FSHβ13AA/FSHβ16AA on days 25 to 28 in the prepubertal male mice promote the complete spermatogenesis on day 29. (**A**) Representative photomicrographs of the testes from the control mice containing only spermatogonia (short arrow) and primary spermatocytes (long arrow). Photomicrographs of the testes from FSHβ13AA- (**B**) or FSHβ16AA-injected (**C**) males containing spermatogenic cells at various stages of development, including spermatogonia (short arrow), primary spermatocyte (long arrow), early spermatids (ellipse), late (mature) spermatids (square). Testes were stained with H&E (×400, scale bar = 10 μm).

**Figure 4 ijms-23-11735-f004:**
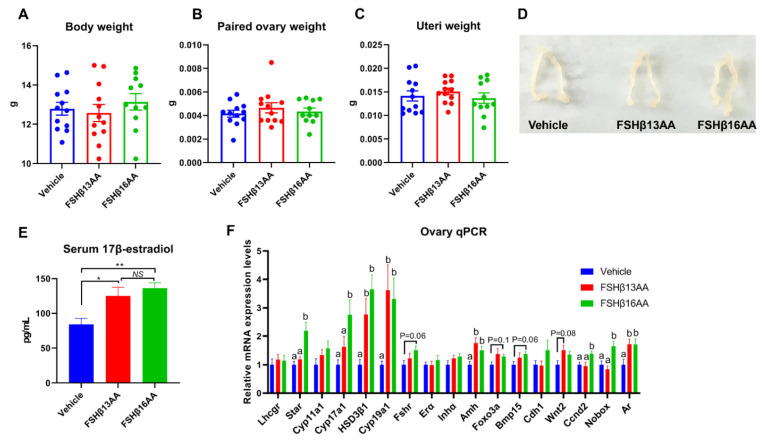
FSHβ13AA/FSHβ16AA on days 25 to 28 in the prepubertal female mice promote ovarian steroidogenesis and the follicle development on day 29. Four consecutive peptide treatments had no significant effects on the body (**A**), paired ovaries (**B**), or uteri (**C**) weights on day 29. (**D**) Representative images of the ovaries and uteri from each group. (**E**) Serum 17β-estradiol concentrations on day 29. * *p* < 0.05, ** *p* < 0.01, NS, not significant. (**F**) mRNA expression levels of the ovarian steroidogenesis- and follicle development-associated genes. *Lhcgr*, luteinizing hormone receptor; *Star*, Steroidogenic acute regulatory protein; *Cyp11a1*, cytochrome P450 family 11 subfamily A member 1; *Hsd3β1*, 3beta-hydroxysteroid dehydrogenase type 1; *Cyp17a1*, cytochrome P450 family 17 subfamily A member 1; *Cyp19a1*, cytochrome P450 family 19 subfamily A member 1 (aromatase); *Fshr*, follicle-stimulating hormone receptor; *Erα*, estrogen receptor 1; *Inha*, inhibin α; *Amh*, anti-Müllerian hormone; *Foxo3a*, forkhead box O3*A*; *Bmp15*, bone morphogenetic protein 15; *Cdh1*, cadherin 1; *Wnt2*, wingless-type MMTV integration site family, member 2; *Ccnd2*, cyclin D2; *Nobox*, NOBOX oogenesis homeobox; *Ar*, androgen receptor. Data are mean ± SEM. ^a,b^ Means without a common letter differed (*p* < 0.05).

**Figure 5 ijms-23-11735-f005:**
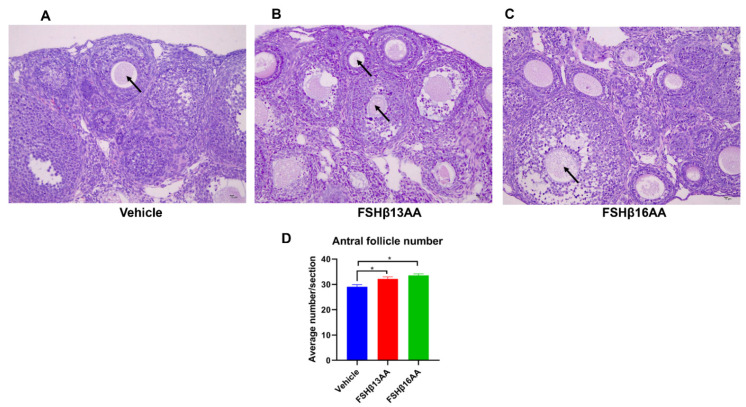
FSHβ13AA/FSHβ16AA on days 25 to 28 promote the ovarian follicular development in prepubertal female mice. Representative images of the antral follicles (arrow) from the vehicle-injected control mice (**A**), FSHβ13AA- (**B**) or FSHβ16AA-injected (**C**) females. (**D**) Morphometric analyses of the antral follicle number from at least three ovaries/group. Tissues were stained with H&E (×200, scale bar = 10 μm). Data are mean ± SEM. * *p* < 0.05.

**Figure 6 ijms-23-11735-f006:**
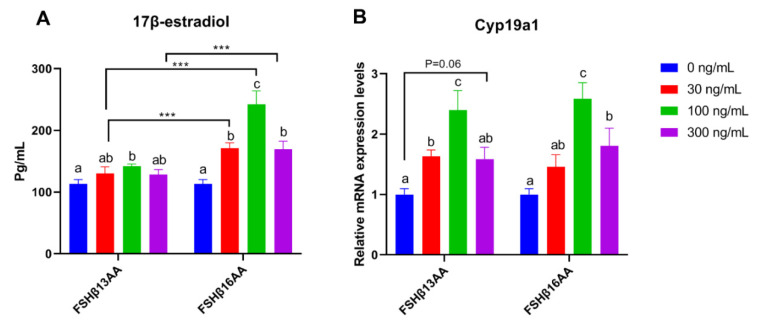
FSHβ13AA/FSHβ16AA promotes the estradiol production in vitro. Effects of the different dosages of FSHβ13AA/FSHβ16AA addition on the 17β-estradiol production (**A**) and the mRNA expressions of *Cyp19a1* (**B**) in the cultured primary ovarian granular cells isolated from the mice on day 25. Data are expressed as mean ± SEM. *Cyp19a1*, cytochrome P450, family 19, subfamily A, member 1 (aromatase). *** *p* < 0.001. ^a–c^ Within a treatment group, the means without a common letter differed.

**Figure 7 ijms-23-11735-f007:**
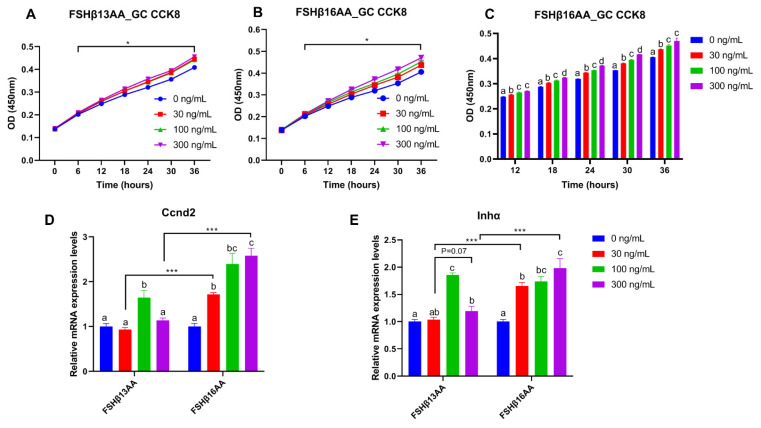
FSHβ13AA/FSHβ16AA promotes the proliferation of the primary murine GCs in vitro. Effects of the dose of FSHβ13AA (**A**) or FSHβ16AA (**B**) on the proliferation of the primary GCs in vitro. * *p* < 0.05 for comparison of the peptide-treated versus the vehicle-treated (0 ng/mL) by repeated-measures 2-way ANOVA with Holm–Sidak’s multiple comparisons test. (**C**) FSHβ16AA promoted the proliferation of the primary ovarian GCs in a dose-dependent manner in vitro. ^a–c^ Within a treatment time point, the means without a common letter differed. (**D**,**E**) Effects of the dose of FSHβ13AA/FSHβ16AA addition on the mRNA expressions of *Ccnd2* and *Inhα* in GCs. *Ccnd2*, cyclin D2; *Inha*, inhibin α. Data are expressed as mean ± SEM. *** *p* < 0.001.

**Figure 8 ijms-23-11735-f008:**
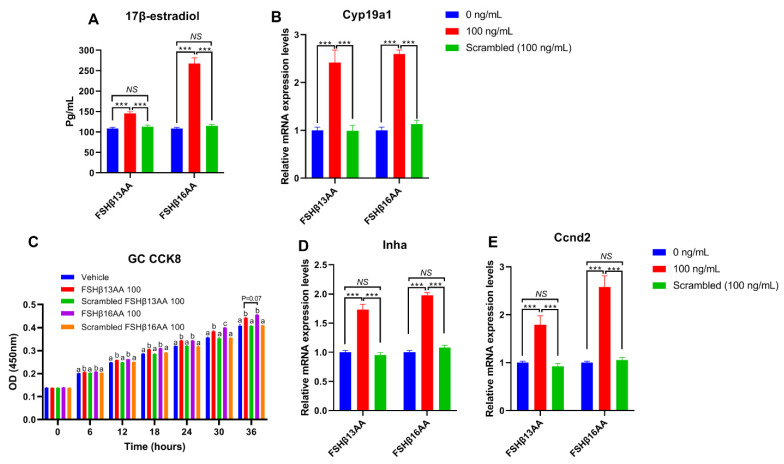
Validating the specificity of FSHβ13AA/FSHβ16AA for inducing the proliferation of primary GCs and estrogen production in vitro. (**A**,**B**) Effects of FSHβ13AA/FSHβ16AA versus their scrambled peptide control (dose = 100 ng/mL) on the 17β-estradiol production (**A**) and the mRNA expressions of *Cyp19a1* (**B**) in the cultured primary ovarian granular cells (GCs). (**C**–**E**) Effects of FSHβ13AA/FSHβ16AA versus their scrambled peptide control on the primary ovarian GCs proliferation (**C**), mRNA expression of *Inha* (**D**) and *Ccnd2* (**E**). Primary ovarian granular cells were isolated from the mice on day 25. Details regarding the scrambled peptides for FSHβ13AA/FSHβ16AA are shown in Appendix A. Data are expressed as mean ± SEM. *Cyp19a1*, cytochrome P450, family 19, subfamily A, member 1 (aromatase), *Ccnd2*, cyclin D2; *Inha*, inhibin α. *** *p* < 0.001; *NS*, not significant (*p* > 0.05); ^a–c^ Within an interval, the means without a common letter differed.

**Figure 9 ijms-23-11735-f009:**
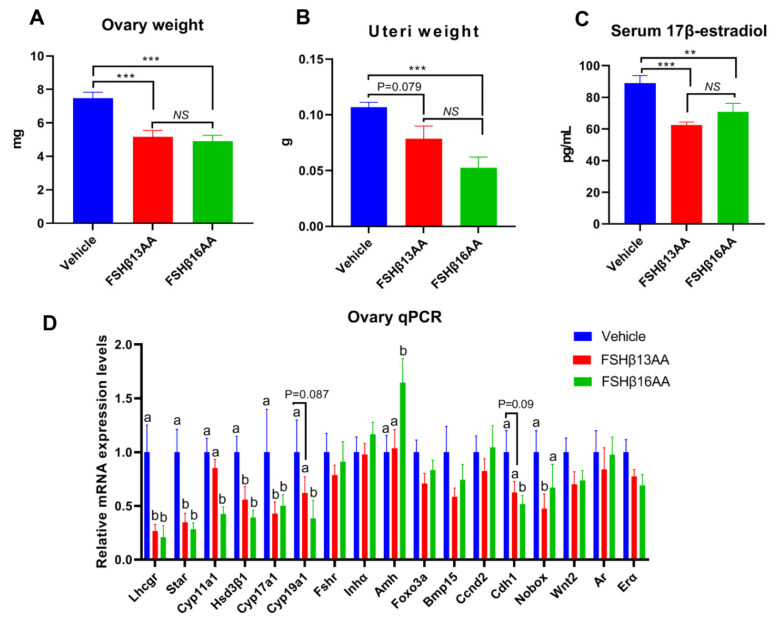
One FSHβ13AA/FSHβ16AA treatment during diestrus antagonizes the FSH-stimulated responses during proestrus in adult female mice. The following data were collected from adult mice during proestrus: (**A**) Paired ovaries weight. (**B**) Uteri weight. (**C**) Serum concentrations of 17β-estradiol. (**D**) Ovarian mRNA expression levels of the steroidogenesis- and follicle development-associated genes. ** *p* < 0.01, *** *p* < 0.001, *NS*, not significant. ^a,b^ Means without a common letter differed (*p* < 0.05). *Lhcgr*, luteinizing hormone receptor; *Star*, Steroidogenic acute regulatory protein; *Cyp11a1*, cytochrome P450, family 11, subfamily A, member 1; *Hsd3β1*, 3beta-hydroxysteroid dehydrogenase type 1; *Cyp17a1*, cytochrome P450, family 17, subfamily A, member 1; *Cyp19a1*, cytochrome P450,family 19, subfamily A, member 1 (aromatase); *Fshr*, follicle-stimulating hormone receptor; *Inha*, inhibin α; *Amh*, anti-Müllerian hormone; *Foxo3a*, forkhead box O3*A*; *Bmp15*, bone morphogenetic protein 15; *Ccnd2*, cyclin D2; *Cdh1*, cadherin 1; *Nobox*, NOBOX oogenesis homeobox; *Wnt2*, wingless-type MMTV integration site family, member 2; *Ar*, androgen receptor; *Erα*, estrogen receptor 1. Data are mean ± SEM.

**Figure 10 ijms-23-11735-f010:**
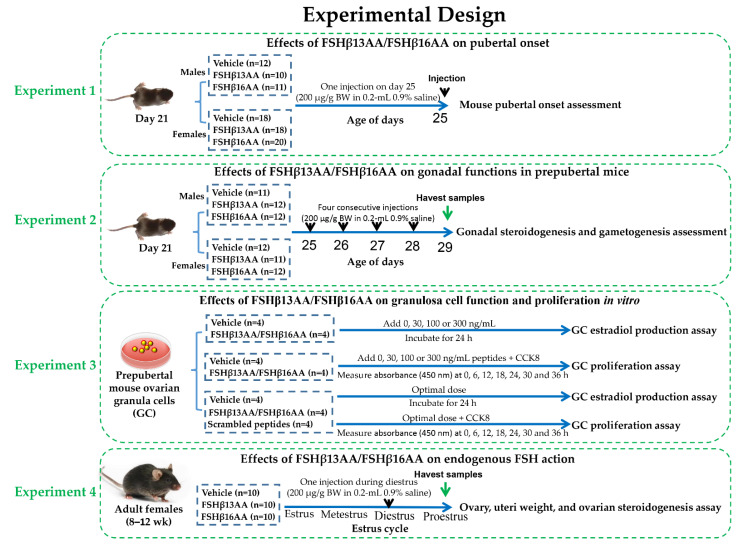
The whole experimental design flow chart.

**Table 1 ijms-23-11735-t001:** qPCR Primer sequences for the tissue genes.

Gene	Genbank Accession No.	Primer Sequence (5′–3′)	Amplification Length (bp)
Fshr	NM_013523.3	F: GTGCATTCAACGGAACCCAG R: TCTAAGCCATGGTTGGGCAG	154
Lhcgr	M81310.1	F: CTGAAAACTCTGCCCTCCAG R: AATCGTAATCCCAGCCACTG	281
Star	NM_011485.5	F: TCCCTCGCAGGACCTTGATCT R: TGGATGGGTCAAGTTCGACG	337
Cyp11a1	AF195119.1	F: AGGTCCTTCAATGAGATCCCTT R: TCCCTGTAAATGGGGCCATAC	137
Hsd3β1	AK147114.1	F: GCGGCTGCTGCACAGGAATA R: GACGCATGCCTGCTTCGTGA	99
Cyp17a1	NM_007809.3	F: GATCGGTTTATGCCTGAGCG R: TCCGAAGGGCAAATAACTGG	81
Cyp19a1	BC103670.1	F: CGGGCTACGTGGATGTGTT R: GAGCTTGCCAGGCGTTAAAG	135
Creb	X67728.1	F: ACTGGCTTGGCACAACCAGA R: GGCAGAAGTCTCTTCATGATT	202
Wnt2	NM_023653.5	F: GCTGAAGTCCTGCTCCTGTG R: CGGTTGTTGTGGAGGTTCAT	183
Klf4	JF277566.1	F: TGGTGCAGCTTGCAGCAGT R: TGGGTTAGCGAGTTGGAAAGG	108
Gja1	XM_036155617.1	F: TTACAACAAGCAAGCCAGCG R: CGTCAGGGAAATCAAACGGC	119
Aqp8	AF018952.1	F: TTGCTACCTTGGGGAACATC R: CCAAATAGCTGGGAGATCCA	121
Erα	AB560752.1	F: GCTCCTAACTTGCTCCTGGAC R: CAGCAACATGTCAAAGATCTCC	75
Inha	NM_010564.5	F:TGAACCAGAGGAGGAAGATGTCTC R:GTCACTGGTCAACTCCAGCAC	82
Amh	NM_007445.3	F:ATCTGGCTGAAGTGATATGG R:CAGGGTATAGCACTAACAGG	105
Foxo3a	NM_001376967.1	F:ACTGAGGAAAGGGGAAATGG R:CAAAGGTGTCAAGCTGTAAACG	123
Bmp15	NM_009757.5	F:GCACGATTGGAGCGAAAATG R:CGTACGCTACCTGGTTTGATGC	123
Cdh1	NM_007988.3	F: TTGGTGTGGGTCAGGAAATC R: GTGTCCCTCCAAATCCGATAC	91
Ar	NM_013476.4	F: GGCAGTCATTCAGTATTCC R: AGTAGAGCATCCTAGAGTTG	89
Ccnd2	XM_036165787.1	F: CAGAAGGACATCCAGCCGTAC R: TCGGGACTCCAGCCAAGAA	136
Nobox	XM_030255220.1	F:CTATCCTGACAGTGACAAACGCC R:CACCCTCTCAGCACCCTCATTAT	331
18S	NR_003278.3	F: TGACTCAACACGGGAAACCT R: AACCAGACAAATCGCTCCAC	125

Abbreviations: *Fshr*, follicle-stimulating hormone receptor; *Lhcgr*, luteinizing hormone receptor; *Star*, Steroidogenic acute regulatory protein; *Cyp11a1*, cytochrome P450, family 11, subfamily A, member 1; *Hsd3β1*, 3beta-hydroxysteroid dehydrogenase type 1; *Cyp17a1*, cytochrome P450, family 17, subfamily A, member 1; *Cyp19a1*, cytochrome P450, family 19, subfamily A, member 1 (aromatase); *Creb*, cAMP response element-binding protein; *Wnt2*, wingless-type MMTV integration site family, member 2; *Klf4*, Kruppel-like factor 4; *Gja1*, gap junction alpha-1 protein; *Aqp8*, aquaporin 8; *Erα*, estrogen receptor 1; *Inha*, inhibin α; *Amh*, anti-müllerian hormone; *Foxo3a*, forkhead box O3*A*; *Bmp15*, bone morphogenetic protein 15; *Cdh1*, cadherin 1; *Ar*, androgen receptor; *Ccnd2*, cyclin D2; *Nobox*, NOBOX oogenesis homeobox; *18s*, 18S ribosomal RNA.

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
