# Peer review of "Two Synthetic Peptides Corresponding to the Human Follicle-Stimulating Hormone β-Subunit Promoted Reproductive Functions in Mice"

_ijms, 2022, doi:10.3390/ijms231911735_

Round 1

Reviewer 1 Report

In this study the authors tested two synthetic peptides corresponding to hFSH-β-(37-49) (LVYKDPARPKIQK, FSHβ13AA) and hFSH-β-(37-49, FSHβ16AA) both identified in the chain of the receptor-binding site of hFSH-β subunit, in modulating reproductive functions in mice. They injected these synthetic peptides in prepubertal male and female mice and observed the hastening of puberty, as well a promoted gonadal steroidogenesis and gamete formation. Similar results were obtained with cultured primary ovarian granular cells treated in the presence oof each or both peptides.

The study is well performed, clearly described and results are well and consistently discussed. Interestingly, results obtained in in vivo were further validated by those obtained in in vitro conditions.

Minor:

Methods: Insert a flow chart to better describe how many samples are have been tested in which group and the kind of experiments carried out.

Other minor observations have been pointed out in the enclosed PDF

Author Response

Responds to the comments of Reviewer #1:

Q1: Methods: Insert a flow chart to better describe how many samples are have been tested in which group and the kind of experiments carried out.

Response: According to the Reviewer’s good suggestion, we have inserted a flow chart (Figure 10) to well describe the sample number and other details in each group of each single experiment in the manuscript.  

Q2: Minor detailed issues/observations that the Reviewer pointed out in the manuscript.

Response: Dear Reviewer, we highly appreciated your patient reviewing on our manuscript. All the the minor issues/mistakes that you pointed out in the manuscript have been well addressed, responded (see ‘Author's Notes to Reviewer#1’, uploaded to the system) and marked in red in the revised manuscript. Please check in the revised manuscript.

Special thanks to your good comments and suggestions!

Reviewer 2 Report

I congratulate the authors to the interesting, well elaborated manuscript which in my opinion has a great meaning for further practical application in human as well as in veterinary medicine! Therefore I highly recomend this manuscript for publication in Int. J. Mol. Sci.

Author Response

Responds to the comments of Reviewer #2:

Dear reviewer, we highly appreciated your positive comments on our research project and manuscript. Thanks again!